# An Optimised Monophasic Faecal Extraction Method for LC-MS Analysis and Its Application in Gastrointestinal Disease

**DOI:** 10.3390/metabo12111110

**Published:** 2022-11-14

**Authors:** Patricia E. Kelly, H Jene Ng, Gillian Farrell, Shona McKirdy, Richard K. Russell, Richard Hansen, Zahra Rattray, Konstantinos Gerasimidis, Nicholas J. W. Rattray

**Affiliations:** 1Strathclyde Institute of Pharmacy and Biomedical Sciences (SIPBS), University of Strathclyde, Glasgow G4 0RE, UK; 2Bacteria, Immunology, Nutrition, Gastroenterology and Omics (BINGO) Group, University of Glasgow, Glasgow G12 8QQ, UK; 3School of Medicine, Dentistry & Nursing, University of Glasgow, Glasgow Royal Infirmary, Glasgow G12 8QQ, UK; 4Royal Hospital for Children and Young People, 50 Little France Crescent, Edinburgh EH16 4TJ, UK; 5Royal Hospital for Children, 1345 Govan Road, Glasgow G52 4TF, UK

**Keywords:** mass spectrometry, metabolite extraction, inflammatory bowel disease, Crohn’s disease, coeliac disease

## Abstract

Liquid chromatography coupled with mass spectrometry (LC-MS) metabolomic approaches are widely used to investigate underlying pathogenesis of gastrointestinal disease and mechanism of action of treatments. However, there is an unmet requirement to assess faecal metabolite extraction methods for large-scale metabolomics studies. Current methods often rely on biphasic extractions using harmful halogenated solvents, making automation and large-scale studies challenging. The present study reports an optimised monophasic faecal extraction protocol that is suitable for untargeted and targeted LC-MS analyses. The impact of several experimental parameters, including sample weight, extraction solvent, cellular disruption method, and sample-to-solvent ratio, were investigated. It is suggested that a 50 mg freeze-dried faecal sample should be used in a methanol extraction (1:20) using bead beating as the means of cell disruption. This is revealed by a significant increase in number of metabolites detected, improved signal intensity, and wide metabolic coverage given by each of the above extraction parameters. Finally, we addressed the applicability of the method on faecal samples from patients with Crohn’s disease (CD) and coeliac disease (CoD), two distinct chronic gastrointestinal diseases involving metabolic perturbations. Untargeted and targeted metabolomic analysis demonstrated the ability of the developed method to detect and stratify metabolites extracted from patient groups and healthy controls (HC), highlighting characteristic changes in the faecal metabolome according to disease. The method developed is, therefore, suitable for the analysis of patients with gastrointestinal disease and can be used to detect and distinguish differences in the metabolomes of CD, CoD, and HC.

## 1. Introduction

Metabolomics is a powerful tool for detecting small molecule cellular and microbial products. Through the reflection of active physiological mechanisms, metabolite characterisation and quantification can give critical insights into human health and disease. The large abundance and diversity of metabolites that are present in human faecal samples, as given by the identification of 6791 faecal metabolites on the Human Metabolome Database (HMBD —https://hmdb.ca/ accessed on 11 November 2022) [1], provides an ideal target for metabolomic analysis [2] and, thus, allows for insights into the outcomes of gut-microbial interactions and dietary impacts on disease [3]. Accumulating evidence indicating the involvement of the gut metabolome in a multitude of diseases [4,5,6] has propelled an intense interest in the role of faecal metabolites under certain environments. The accurate quantification of metabolites in faecal samples, therefore, holds value in a wide range of research areas. A clear role of faecal metabolomics has been demonstrated in the field of gastrointestinal disease, including inflammatory bowel disease (IBD) [7] and coeliac disease (CoD) [8]. Although the aetiology of such diseases remains elusive, shifts in metabolic profile are associated with disease activity and may represent central components of pathogenesis [9,10,11,12]. Irrespective, detection of altered patient metabolites may help unravel underlying disease mechanisms or reveal new diagnostic or prognostic markers of clinical utility.

Liquid-chromatography mass spectrometry (LC-MS) is a popular metabolite analysis technique due to its high sensitivity and selectivity. Sample preparation and pre-treatment is a vital stage of the LC-MS workflow, providing the scaffolding to support metabolite detection. The experimental framework therefore shapes the biological interpretation of a metabolomics study, and so it is crucial to consider best practices regarding specific study aims. Certain challenges are inherent in the sample preparation phase, such as the large physio-chemical variation of the target metabolite pool, technical and environmental variation, and the complex and heterogeneous nature of human faeces. This brings difficulties in standardising metabolomic methods, which is evident in the lack of “gold standard” metabolite extraction procedures. As the effective and reliable identification of metabolites is largely dependent on the extraction method used, it is imperative to consider sample preparation when comparing results between studies. To date, previous studies have addressed some of the challenges associated with metabolomic sample preparation [13,14,15]; however, these are mainly based on biphasic extraction protocols with limitations in scalability. While efficient biphasic extraction systems for faecal analysis contribute towards protocol standardisation, they are associated with complicated handling due to the requirement for phase separation. It can, therefore, be challenging to utilise two-phase protocols in large scale clinical studies, with further limitations in protocol automation. With the increasing demand for translating metabolomics data into meaningful clinical output, one major requirement for bridging the bench to bedside gap is the use of large population studies. It is, therefore, also important to optimise less-complex monophasic extraction protocols that can be used as an alternative to classical biphasic protocols for LC-MS analysis. Moreover, the applicability of metabolite extraction in the context of gastrointestinal disease requires further acknowledgement. Thus, the present study has the goal of advancing a method for monophasic metabolite extraction that can be easily implemented in large scale clinical studies investigating gastrointestinal disease. To the best of our knowledge, there is no current documentation on optimal extraction methods for IBD or CoD samples for LC-MS analysis. There is an important unmet requirement for the effect of faecal sample type to be explored, which is exemplified here in the comparison between gastrointestinal disease and the non-disease state.

Herein, we evaluate different faecal extraction methods for metabolomic measurements in human faecal samples from healthy individuals, Crohn’s Disease (CD) and CoD patients. A range of trial experiments were performed to determine the optimal sample weight, extraction solvent, disruption method, and sample-to-solvent ratio using LC-MS. The overall aim of this study is twofold; firstly, to optimise metabolite extraction parameters for faecal samples and secondly, to determine whether this optimised extraction protocol is suitable for analysis of samples from patients with gastrointestinal disease. To capture the large quantity of metabolites and ensure maximal coverage in the method development phase, untargeted metabolomic analysis was performed to assess each sample parameter. Targeted metabolomic analysis was subsequently applied to assess method suitability in patients with disease.

## 2. Materials and Methods

### 2.1. Ethics Statement

All participants and their carers provided written informed consent. The study was approved by the NHS West of Scotland Research Ethics Committee (14/WS/1004 for Crohn’s disease patients and 11/WS/0006 for patients with coeliac disease).

### 2.2. Faecal Samples

Faecal samples were collected for metabolomic analysis within 2 h of passage, kept in anaerobic conditions (Anaerocult™ A) and inside an ice box with ice packs. The samples were transferred to the laboratory immediately, homogenised with mechanical kneading, and aliquots were kept at −80 °C until further processing. After metabolite extraction, the samples were again kept at −80 °C until LC-MS analysis. The samples were kept on ice during transportation.

### 2.3. Chemicals and Reagents

LC-MS grade methanol (MeOH), acetonitrile (ACN), chloroform (CHCl_3_), and water (H_2_O) were purchased from Fisher Scientific (Geel, Belgium). LC-MS grade formic acid was purchased from Thermo Scientific (Prague, Czech Republic).

### 2.4. Extraction Protocol

Freeze-dried faecal samples were added to the extraction solvent and the cells were disrupted using bead beating (FastPrep 24 MP Biomedicals), sonication, and freeze-thaw lysis methods. The samples were then centrifuged at 13,000× *g* for 15 min and the supernatant recovered. The samples were evaporated to dryness using a SpeedVac Savant SPD121P system (Thermo Scientific, Milford, UK) and stored at −80 °C until further processing. Reconstitution was performed in 250 μL 50/50 H_2_O: acetonitrile (ACN), vortexed for 1 min and centrifuged at 15,000× *g* for 15 min, and aliquots transferred into glass vials for MS analysis. Quality control (QC) samples were prepared by pooling samples across all groups undergoing simultaneous analysis. Solvent blanks and QC samples were entered at the beginning of every analytical run, and after every five samples in each batch over the course of the study to assess background in the system and detect potential contaminations. Experimental details for each extraction parameter are shown (Table 1). 

### 2.5. Untargeted LC-MS Metabolite Measurement

Untargeted metabolomic analysis was performed on an ultra-high performance liquid chromatography (UHPLC) system (ThermoFisher Scientific) coupled to an Orbitrap Exploris 240 (ThermoFisher Scientific) mass spectrometer. The LC-MS method was previously optimised on the Orbitrap system, with the settings transferred from the applied method [16]. Chromatographic separation was performed on a Vanquish Accucore C18 + UHPLC analytical column (ThermoScientific, 100 mm × 2.1 mm, 2.6 μM) at a flow rate of 400 μL min^−1^. Mobile phase A was composed of 99.9% water + 0.1% formic acid and mobile B was composed of 99.9% MeOH + 0.1% formic acid. Electrospray ionisation (ESI) was used as the ionisation method, set at 3900 V and 2500 V for positive and negative mode, respectively. The elution gradient used can be found in Appendix A. The source-dependent parameters were operated under the following conditions: sheath gas, 40 Arb; auxiliary gas, 10 Arb; sweep gas, 1 Arb; ion transfer tube temperature, 300 °C; vaporiser temperature, 280 °C. Instrument calibration was performed using Pierce^TM^ FlexMix^TM^ calibration solution (Thermo Scientific) and ran under vendor recommended settings. MS data collection was performed in a top-5 data dependent acquisition mode (DDA) to give putative MSMS metabolite identification at MSI level 2.

### 2.6. Targeted LC-MS Metabolite Measurement

Targeted metabolomic analysis was performed on a UHPLC system coupled to a triple quadrupole mass spectrometer (Shimadzu 8060NX, Kyoto, Japan). The method used for metabolite detection and quantification was provided by the vendor; Primary Metabolites LC/MS/MS Method Package version 2.0 (Shimadzu, Kyoto, Japan). The method was designed to detect 97 metabolites. The list of 97 detected metabolites and associated parameters are shown in Appendix A. Chromatographic separation was performed on a pentafluorophenylpropyl (PFPP) + UHPLC analytical column (Merck, 150 mm × 2.1 mm, 3 μM) at a flow rate of 400 μL min^−1^. Mobile phase A was composed of 99.9% water + 0.1% formic acid and mobile B was composed of 99.9% acetonitrile + 0.1% formic acid. Electrospray ionisation (ESI) was used as the ionisation method, set at 3900 V and 2500 V for positive and negative mode, respectively. The source-dependent parameters were operated under the following conditions: column oven temperature, 40 °C, nebulising gas flow rate, 3.0 L min^−1^, drying gas flow rate, 10 L min^−1^, desolvation line temperature, 250 °C, and block heater temperature, 400 °C. The elution gradient used can be found in Appendix A.

### 2.7. Method Application

We applied the method to three biological groups: CD patients, CoD patients, and HCs (Table 2). CD patients were undergoing varying forms of treatment and CoD patients were following a gluten-free diet. HCs were defined as individuals with the absence of gastrointestinal disease. Both untargeted and targeted metabolomic analyses were applied to the sample sets combined after randomisation.

### 2.8. Mass Spectrometry Data Processing

For the processing of untargeted metabolomics data, Thermo Scientific Xcalibur format raw data files (.RAW) were imported into Compound Discoverer software version 3.2 (Thermo Fisher Scientific, Waltham, MA, USA). Details of the workflow for analysis in Compound Discoverer is included in Appendix A. The targeted metabolomics data were converted from Shimazdu vendor format (.lcd) to mzML format. A data matrix of identified metabolites and associated peak areas was constructed and processed using R-Studio v 3.5.2 (RStudio, PBC MA, USA). 

### 2.9. Data and Statistical Analysis

For untargeted analysis, principal component analysis (PCA) was performed using Compound Discoverer software 3.2 (Thermo Fisher Scientific, Waltham, MA, USA). For targeted analysis, PCA was performed using Lab Profiling Solutions software version 5.6 (Shimadzu, Kyoto, Japan) and R-Studio (RStudio, PBC, MA, USA). PCs were calculated using prcomp function and PCA scores plots were generated using the following packages in R: ggplot2, ggfortify, grid, and gridExtra. Differential analysis using volcano plots allowed significant differences between groups to be determined. Univariate statistical analyses were performed using unpaired *t*-test and one-way ANOVA, with the level of significance set at *p* < 0.05. Central network analysis was performed in R-studio (RStudio, PBC, MA, USA). using the igraph package.

### 2.10. Putative Metabolite Identification

The inclusion criteria for putative metabolite identification were set and applied to refine the total number of features. After removing duplicates, the resulting list contained only unique metabolites within a 4 ppm mass accuracy range and with a full mzCloud match. Contaminations were excluded by analysing 50:50 H_2_O/MeOH blank samples throughout the MS run. Metabolite annotation was performed manually and using the HMDB.

## 3. Results

For method development, metabolites were measured in freeze-dried faecal samples obtained from healthy participants. The metabolic output was first measured by PCA to observe any differences in the overall metabolic signature obtained from each method. Further statistical analysis was performed for data quantification by calculating the number of *m*/*z* features, putatively identified metabolites, signal intensity, and metabolic coverage.

### 3.1. Analysis of Sample Weight

Positive ionisation mode was used for analysis of experimental parameters as previous investigations found that a significantly higher number of *m*/*z* features were detected in comparison to the negative ionisation mode. While examining the effect of sample weight, 10 mg samples were disregarded during the extraction process as the aliquots had very little extractable supernatant for subsequent processing. This was likely due to the sample being absorbed by the zirconium beads as the sample weight was too small for the solvent volume. During the reconstitution step, the 100 mg sample was also disregarded as there was too much particulate left undissolved. The metabolites were successfully extracted from 20 mg and 50 mg samples and measured using LC-MS and PCA demonstrated clear separation of the two sample weight groups (Figure 1). In this case, 50 mg samples showed a significantly higher mean number of *m*/*z* features and mean number of putatively identified metabolites in comparison to 20 mg samples. Furthermore, the mean signal intensity given by 50 mg samples (2.1 × 10^7^) was significantly increased compared to 20 mg samples (1.2 × 10^7^). It was furthermore demonstrated that 69.1% of detected metabolites were found at significantly increased levels in 50 mg samples compared to 20 mg samples (Appendix A). A comparison of the total number of metabolites per chemical class from each sample weight is shown in Appendix A.

### 3.2. Analysis of Extraction Solvent

PCA demonstrated a clear separation of the extraction solvents (Figure 2). Using 100% MeOH gave a significantly higher number of *m*/*z* features in comparison to 1:1 MeOH/H_2_O and a significantly higher number of putatively identified metabolites than both MeOH/H_2_O and 2:1 CHCl_3_/MeOH. No significant differences were observed in the signal intensity between the extraction solvents. Differential analysis revealed a significant increase in the levels of 30.6% and 20.9% of metabolites detected using MeOH as the extraction solvent in comparison to MeOH/H_2_O and CHCl_3_/MeOH, respectively (Appendix A). In this case, 32.0% of metabolites detected were found at significantly increased levels in CHCl_3_/MeOH compared to MeOH/H_2_O. MeOH extractions additionally had a significantly increased number of lipids compared to MeOH/H_2_O extractions. Once more, all metabolite classes were detected from all extraction solvents, with a similar structure of metabolite classification. A comparison of the total number of metabolites per chemical class from each extraction solvent is shown in Appendix A.

### 3.3. Analysis of the Cellular Disruption Method

The choice of cellular disruption method affected the overall metabolic output, as shown by PCA which demonstrated a clear separation between the three groups (Figure 3). Bead beating extracted a significantly higher mean number of *m*/*z* features in comparison to freeze-thawing and a significantly higher number of putatively identified metabolites than both sonication and freeze-thawing. No significant differences were observed in the signal intensity between lysis methods. A significant increase in the levels of 29.5% and 48.4% of metabolites detected were found using bead beating as the method of cellular disruption compared to sonication and freeze-thawing, respectively (Appendix A). Of the metabolites identified, 23.7% were found at significantly increased levels in sonicated samples in comparison to freeze-thawing. Each disruption method allowed for the measurement of metabolites from all classification groups. While similar patterns of metabolite classification are shown between methods, it was shown that bead beating led to detection of a significantly increased number of lipids compared to the other lysis techniques. A comparison of the total number of metabolites per chemical class using each cellular disruption method is shown in Appendix A.

### 3.4. Analysis of Sample-to Solvent Ratio

A clear separation was observed between the three different sample-solvent ratios by PCA (Figure 4). Performing extractions using a ratio of 1:20 gave a significantly higher mean number of *m*/*z* features and putatively identified metabolites than ratios of 1:5 and 1:10. Furthermore, a significant increase in the signal intensity of samples of a 1:20 ratio was observed in comparison to the other groups. A significant increase in the levels of 70.0% and 66.7% of metabolites detected were found using a ratio of 1:20 in comparison to ratios of 1:5 and 1:10, respectively (Appendix A). In this case, 43.5% of metabolites detected were found at significantly increased levels in samples extracted using a ratio of 1:10 compared to 1:5. Several metabolite classes were increased in extractions carried out using a ratio of 1:20 compared to the other groups. Additionally, the overall composition according to chemical class of each sample remained similar between each group. A comparison of the total number of metabolites per chemical class using each sample-to-solvent ratio is shown in Appendix A.

Through the exploration of the overall metabolite extraction efficiency through the optimisation process, it was observed that the number of putatively identified metabolites significantly increased throughout stages of method optimisation with the improvement of each individual extraction parameter (Appendix A).

### 3.5. Applicability of the Method to Patients with Gastrointestinal Disease

To assess the applicability of the developed method, we applied the protocol to CD, CoD, and HC groups and compared the metabolic differences. In an untargeted analysis, PCA demonstrated a clear separation between CD samples and the other groups (Figure 5). A significant decrease in the levels of 72.3% of metabolites detected were found in CD samples compared to HCs, and 74.1% compared to CoD samples (Appendix A). Of the metabolites detected, 27.1% were found to be at significantly decreased levels in CoD samples in comparison to HCs. Furthermore, targeted metabolomics analysis further confirmed the ability of the method to both detect and stratify metabolites extracted from faecal sample from patients with CD and CoD and healthy individuals. PCA showed characteristic changes in the faecal metabolome between each of the groups (Figure 6). In order to ensure the present method was effective in the specific context of gastrointestinal disease, we carried out further analysis investigating metabolites that are important in IBD. The metabolites that were putatively identified in the current method and throughout the literature in the context of IBD are compared (Appendix A).

## 4. Discussion

Since extraction methodology directly affects metabolite constitution within MS metabolomics experiments, it was important to optimise a range of experimental parameters and to document the chemical coverage in faecal samples. To this end, the present study first aimed to assess parameters of maximal metabolic LC-MS output, utilising an untargeted metabolomics approach to allow fingerprinting of the total metabolite profile in samples. The ideal extraction protocol was therefore one that elucidated the greatest number of metabolites whilst minimising interferences. As such, methods were evaluated by measuring the total number of metabolites detected using each protocol. Since we cannot assume that the number of features is equal to the number of correctly identified metabolites, due to unmatched features, blanks, and duplicate readings, further refinement methods were applied to allow for a more accurate evaluation of the protocols. Additionally, the markedly different characteristics of metabolites in the faecal metabolite pool brings challenges in extracting all the metabolites present in each sample. For this reason, it was important to assess the number of metabolites belonging to different metabolic classes from each method to ensure maximum chemical coverage. Feature annotation was performed to quantify and compare metabolite classifications between the extraction methods. As a complete characterisation of the metabolome is not possible, a compromise will always exist in practice; however, the multi-parameter method used in the present study allows for the selection of the greatest metabolite signal and coverage.

Herein, we describe an optimised protocol for extraction of metabolites from human faecal samples, thus providing an efficient setup for subsequent metabolomic analysis. The method is recapitulated in the following stages: (1) 50 mg sample weighed out, (2) 1000 μL MeOH added to sample and cell lysed by bead beating, (3) samples evaporated to dryness under vacuum and stored at −80 °C until further processing, (4) reconstitution carried out in 50/50 ACN: H_2_O, (5) LC-MS analysis using 1 μL injection volume (Appendix A).

The metabolite extraction from 10 mg and 100 mg samples were unsuitable for metabolomic analysis and therefore not included in the results. This is important, as when run on the MS, sample particulate may crash the column and lead to instrument breakdown. The faeces weight-to-solvent ratio (100 μL of solvent for every 10 mg of sample) was, therefore, not sufficient for samples out with a 20–50 mg range. For this reason, we explored the impact of sample-solvent ratio on metabolic output in a further analysis. In consideration to this, for the assessment of sample weight, 20 mg and 50 mg samples were successfully extracted and metabolomic analysis was continued. A clear separation was shown by the PCA comparing 20 mg and 50 mg samples, indicating the different metabolite profiles given by the two groups. Further analysis showed that 50 mg samples additionally contained an increased number of *m*/*z* features, identified metabolites, and signal intensity—this result was to be expected due to the increased levels of biomass in the 50 mg samples. It was also important to investigate whether the observed differences in metabolite numbers were reflected in the overall metabolic coverage. Thus, the detected metabolites were grouped according to their chemical classification, and calculation of the number of metabolites in each class was used as a measurement of metabolic coverage. This is essential for untargeted metabolomics experiments, as the analytical conditions should aim to detect a broad range of metabolites of different chemical properties that may be implicated in disease. As such, expansion of metabolic coverage is important to maximise information for hypothesis generation. From the classification analysis, it was revealed that metabolite class is conserved across sample weight. Using 50 mg faecal samples for metabolite extraction aligns with previously reported studies [3,17,18,19], in which 50 mg samples were also used as the starting point for sample preparation and subsequent analysis. Based on findings of increased metabolite numbers without compromising metabolic coverage or signal intensity, it is reasonable to suggest that 50 mg samples are optimal for use in faecal extraction protocols.

While investigation into extraction solvent was here carried out using MeOH, MeOH/H_2_O, and CHCl_3_/MeOH, it is worth mentioning that other solvents, such as ACN and isopropanol have previously been used in faecal extractions. However, due to limited clinical sample availability, the extraction solvents for this study were chosen based on a previous literature search. The results from this analysis showed a clear separation between protocols using MeOH, MeOH/H_2_O, and CHCl_3_/MeOH, with an increased number of *m*/*z* features and identified metabolites given by pure MeOH extractions. While it was shown that the number of lipids and derivatives were increased in the samples extracted using MeOH in comparison to the other groups, the overall metabolic coverage was very similar for all extraction solvents investigated. As maximal chemical coverage is largely maintained, it can again be noted that metabolite class is conserved across extraction solvents. As the use of pure MeOH increases overall metabolic features obtained from molecules across a wide range of different chemical properties, its use can therefore be recommended as the optimal solvent for faecal extraction. This result agrees with a recently reported study, where MeOH was chosen as the optimal solvent for the extraction of metabolites from human faecal samples in order to assess gut health [20]. Furthermore, MeOH has been found to be the optimal extraction solvent in a range of metabolomics studies, including the investigation of dietary influences in faecal samples [3], serum metabolite profiling [21], and *Blastocystis’* metabolism [22]. In comparison with one of the current most used extraction solvents, phosphate buffer saline (PBS) [23], the recognition of MeOH as an efficient organic buffer and resultant choice in a range of sample preparation methods may be attributed to effective protein denaturation [24] and multi-polarity chemical capture [25].

Cell lysis is the process of breaking down the cell membrane to release contents contained inside the cell for molecular analysis. Bead beating, sonication, and cycles of freeze-thawing are common techniques used to disrupt the cell, and a sense of uncertainty resides about optimal methodological choice. The samples that underwent cell lysis using bead beating contained a significantly higher number of *m*/*z* features than freeze-thawing and a significantly increased number of identified metabolites than both those with sonication and freeze-thawing. Moreover, cell disruption by bead beating had a significantly increased number of lipids compared to both other methods. Overall, these findings indicate that bead beating was the most effective cell lysis method for extracting metabolites from human faecal samples. Additional studies have found analogous findings; for example, one study showed that bead beating was the best method for cell disruption and subsequent extraction of both polar and non-polar compounds from platelet samples, as given by optimal extraction efficiencies [26]. Bead beating has also previously been used as the cell lysis method of choice in the sample preparation of human faecal samples [27], as well as for gastrointestinal stromal tumour [28] and the characterisation of tissue samples [29].

Sample-to-solvent ratio, as aforementioned, is vital not only to maximise the data obtained, but also to ensure sufficient sample quality for LC-MS analysis so as not to cause blockage and instrument breakdown. This is particularly important for complex biomatrices such as faeces, which are composed of an abundance of organic and cellular material. The sample-to-solvent ratio, therefore, must allow extraction of large metabolite numbers that are compatible with LC-MS systems. Therefore, the metabolic output resulting from sample-to-solvent ratios of 1:5, 1:10, and 1:20 were assessed. Different metabolite quantification analyses identified a higher number of *m*/*z* features, identified metabolites, and signal intensity were given by samples using a sample-solvent ratio of 1:20 compared to the other tested ratios. Over 300 *m*/*z* features were detected and putatively identified using the optimal procedure with a 1:20 sample-solvent ratio, which holds great promise for maximising capture of biological information in future metabolomic studies. It is important to note that this work is part of an ongoing effort to document the metabolites putatively identified in faecal samples, which will in future will be built upon by the creation of a standards library and the additional use of pure standards. Putative metabolite identification at MSI level 2 without the use of internal standards is, however, a current limitation of the present study, and the resulting lack of validation techniques must also be highlighted. Nonetheless, this work algins with the reporting standards of chemical analysis [30] and will be extended in future in order to increase the confidence of identification and validity of findings.

While contradictive reports are found regarding metabolite extraction procedures, it is important to bring to light methods that are suitable in specific contexts to continue the drive towards standardisation. The use of biphasic extraction protocols is common in metabolomics sample preparation; however, method advancement must also reflect amenability to study design. A considerable amount of research [29,31,32,33] suggests the importance of single-phase extraction procedures that can be used as simple, fast, and scalable alternatives to some of the more extensive approaches, giving impetus for investigating the optimal monophasic extraction protocol for human faecal samples. Rapid and easy-to-use methods can greatly simplify metabolite extraction and thereby improve scalability and application in large clinical studies. In this sense, single-phase methods are advantageous as the single layer can easily be removed, minimising the risk of sample loss and contamination [34,35]. This is of paramount importance for large studies as well as those with limited sample amounts. Moreover, the method developed in this study uses fewer toxic chemicals and can, therefore, be deemed as more friendly to both the operator and environment [36,37]. However, it must also be noted that while monophasic protocols provide simple and scalable extractions, consideration must also be given to the potential trade-off regarding metabolome coverage in comparison to biphasic methods. Improvements to the automation and scaling of extraction methods for large studies using monophasic methods should be conducted without significantly reducing the metabolome coverage. Extraction methods utilising biphasic partitioning are advantageous in their ability to separately recover polar and non-polar metabolites, ensuring coverage across the polarity scale. While contradictive reports have previously been noted regarding the comparative coverage of monophasic and biphasic approaches [13,37], recent research has provided evidence to suggest the differences in coverage between the two approaches may not be significant. For example, recent studies have demonstrated that single-step sample preparation methods showed metabolome coverage and signal intensities equivalent to or greater than biphasic methods [33,38,39]. Careful consideration is required when implementing metabolite extraction methods to fit the specific study aims; however, in addressing the requirement for simple and rapid extraction methods for large-scale studies, it can be suggested that monophasic methods may be implemented as the best compromise for both scalability and coverage.

Finally, we demonstrated the applicability of the method on samples from patients with two forms of gastrointestinal disease involving metabolic and microbial perturbation, CD and CoD [40,41]. The developed method successfully detected and differentiated metabolic patterns of each group with a wide coverage. The method demonstrates strong cross-platform compatibility, with successful method application using two distinct analytical platforms, Orbitrap 240 LC-MS (ThermoFisher Scientific) and the 8060NX targeted triple-quadrupole (Shimadzu, Kyoto, Japan). This is valuable for future use of the method in laboratories using different technologies for metabolomic analysis.

In summary, the untargeted and targeted LC-MS analysis of different extraction factors provide insights into specific methods which give the strongest metabolic output. Optimised sample pre-treatment and extraction methods ultimately improve protocol efficiency while simultaneously enhancing the MS signal obtained [42]. Each small parameter change may cause a small increase in the efficiency of LC-MS characteristics and so when combined, the accumulated difference in the overall protocol can result in a large improvement to the number and coverage of metabolites detected (Appendix A). Furthermore, reproducibility of the method and the instrument are increased by documenting and working towards method standardisation. As the results from this study bring together some of the parameters of faecal metabolite extraction in agreement with existing studies, this supports evidence of an optimised and reproducible protocol that can be applied in a vast array of research and clinical settings. Moreover, the method covers a wide range of metabolites of different physiochemical properties to increase the capture of biological compounds. As an extension, employing the method to patients with gastrointestinal disease expands the protocol applicability to different sample types. This method addresses the requirement for affordable, reproducible, and environmentally friendly metabolite extraction protocols. Thus, the method described build on the foundations of protocol standardisation, allowing for improved comparisons of future metabolomics studies using faecal samples.

## 5. Conclusions

Based on a series of optimisation experiments, we describe a protocol to extract metabolites from faecal samples for metabolomic analysis using an LC-MS system. We recommend the use of 50 mg freeze-dried faecal samples in a 1000 μL MeOH and bead beating extraction, as given by a reproducible increased metabolite measurement. The optimised faecal extraction method described here can be used for metabolomics investigations of a wide array of applications, with strong evidence for its suitability in studies of gastrointestinal disease. This contributes towards standardising a framework of sample preparation, allowing for easier and more accurate comparisons between studies.

## Figures and Tables

**Figure 1 metabolites-12-01110-f001:**
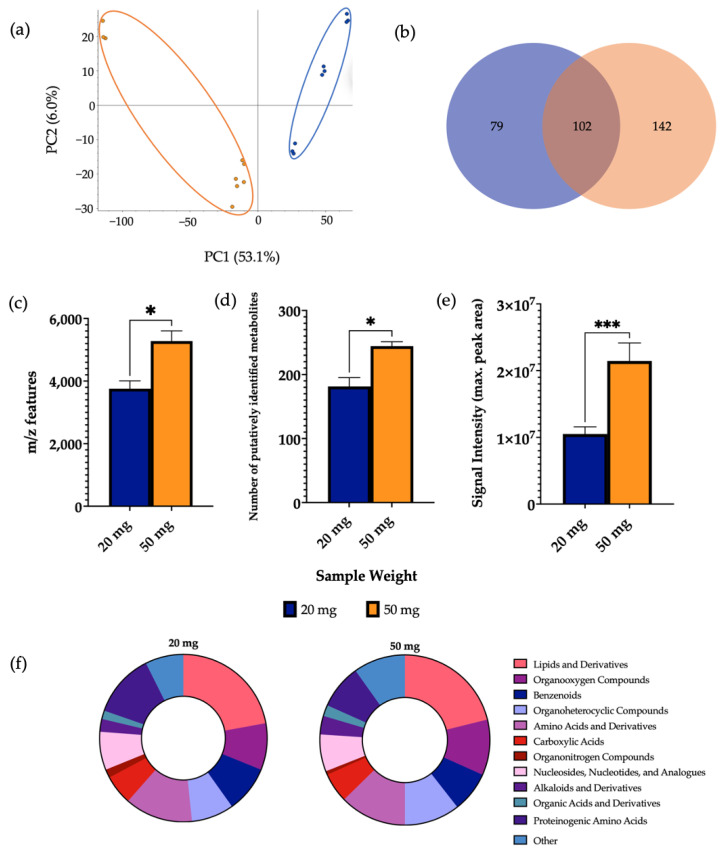
The effect of sample weight on features of metabolomic analysis. 1 μL of 20 mg and 50 mg sample was injected onto a C18 column (*n* = 3), performed in triplicate. (**a**) PCA of metabolomic profiles obtained as a function of sample weight. PCA score plots demonstrating extracted faecal metabolites between different sample weights. Discrimination between 20 mg (blue) and 50 mg (orange) samples was characterised by a variability of 53.1%. (**b**) A Venn diagram of the mean number of metabolites detected between each method. (**c**) The total number of *m*/*z* features and (**d**) total number of putatively identified metabolites were calculated in positive ionisation mode and (**e**) the overall mean signal intensity of each sample weight was assessed. (**f**) A metabolite class quantification demonstrating the faecal metabolome patterns according to chemical class in 20 mg and 50 mg samples. The bar chart data were expressed as mean ± SEM and statistical significance was assessed using an unpaired *t*-test. * *p* < 0.05, *** *p* < 0.001.

**Figure 2 metabolites-12-01110-f002:**
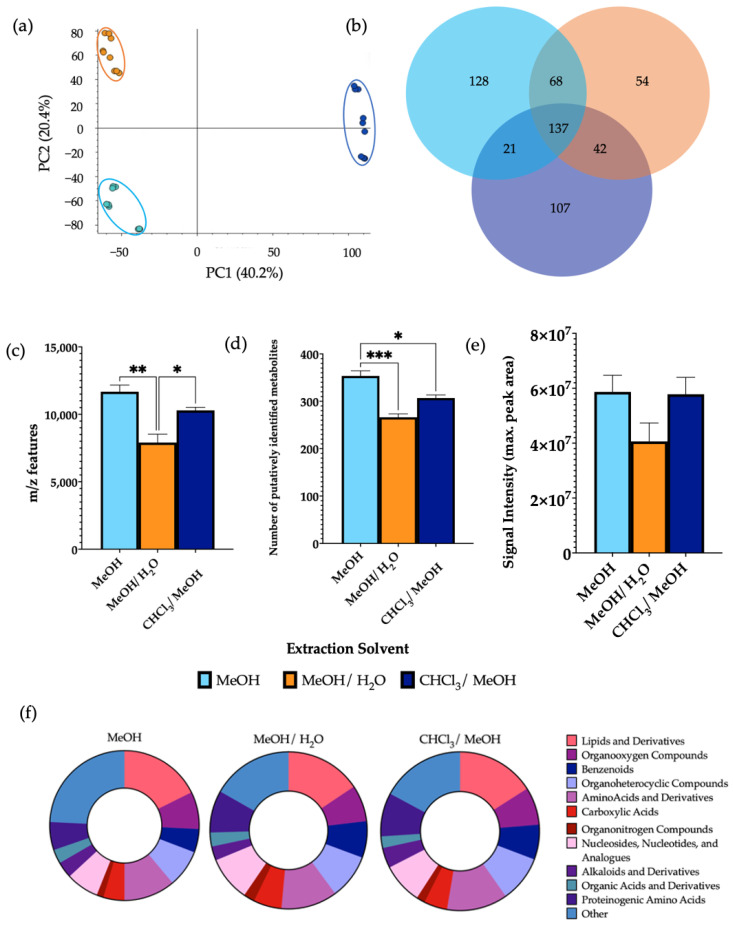
The effect of extraction solvents, MeOH, MeOH/H_2_O, and CHCl_3_/MeOH, on features of metabolomic analysis. 1 μL of each extraction sample was injected onto a C18 column (*n* = 3), performed in triplicate. (**a**) PCA of metabolomic profiles obtained as a function of extraction solvent. PCA score plots demonstrating extracted faecal metabolites between different extraction solvents. Discrimination between extraction solvents MeOH (light blue), MeOH/H_2_O (orange), and CHCl_3_/MeOH (dark blue) was characterised by a variability of 40.2%. (**b**) A Venn diagram of the mean number of metabolites detected between each method. (**c**) The total number of *m*/*z* features and (**d**) total number of putatively identified metabolites were calculated in positive ionisation mode and (**e**) the overall mean signal intensity of each extraction solvent was assessed. (**f**) The metabolite class quantification demonstrating the faecal metabolome patterns according to chemical class in each extraction sample. The bar chart data were expressed as mean ± SEM and statistical significance was assessed using one-way ANOVA. * *p* < 0.05, ** *p* < 0.01, *** *p* < 0.001.

**Figure 3 metabolites-12-01110-f003:**
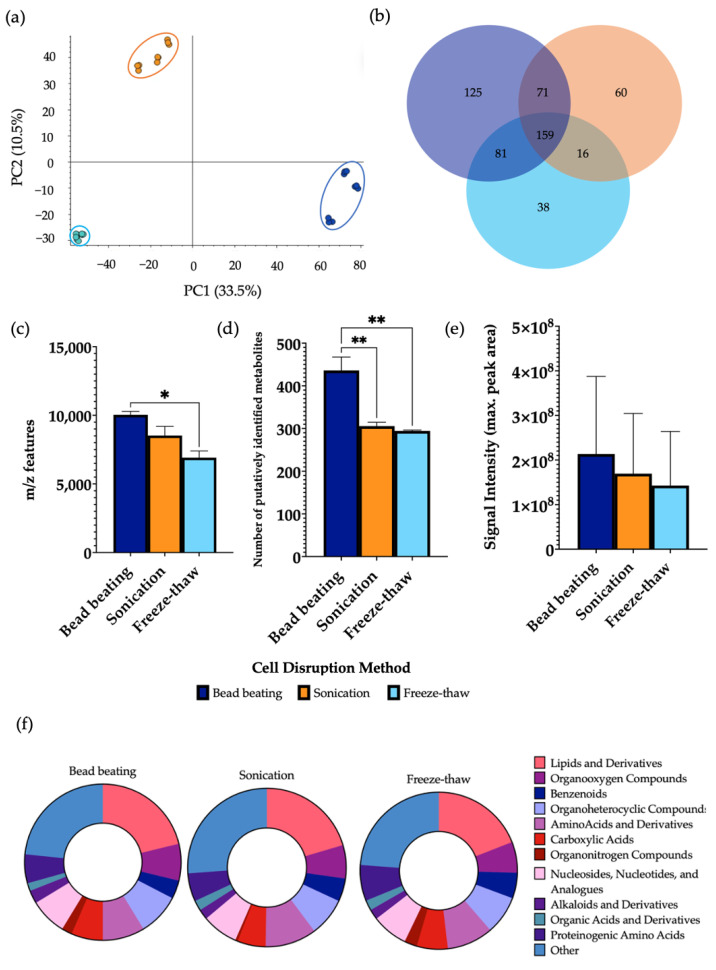
The effect of cellular disruption methods, bead beating, sonication, and freeze-thaw cycles, on features of metabolomic analysis. 1 μL of each extraction sample was injected onto a C18 column (*n* = 3), performed in triplicate. (**a**) PCA of metabolomic profiles obtained as a function of disruption method. PCA score plots demonstrating extracted faecal metabolites between bead beating, sonication, and freeze-thaw cycles. Discrimination between extraction solvents A, bead beating (dark blue); B, sonication (orange) and C, freeze-thaw cycles (light blue) was characterised by a variability of 33.5%. (**b**) A Venn diagram of the mean number of metabolites detected between each method. (**c**) The total number of *m*/*z* features and (**d**) total number of putatively identified metabolites were calculated in positive ionisation mode and (**e**) the overall mean signal intensity of each disruption method was assessed. (**f**) The metabolite class quantification demonstrating the faecal metabolome patterns according to chemical class in each extraction sample. The bar chart data were expressed as mean ± SEM and statistical significance was assessed using a one-way ANOVA. * *p* < 0.05, ** *p* < 0.01.

**Figure 4 metabolites-12-01110-f004:**
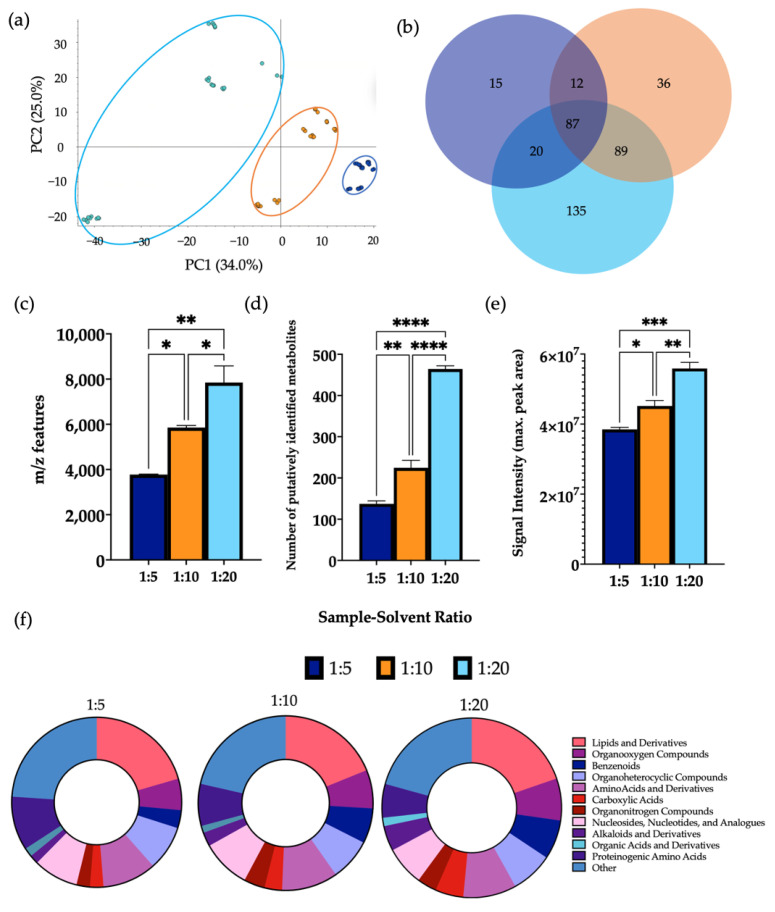
The effect of sample-solvent ratio on features of metabolomic analysis. 1 μL of each extraction sample was injected onto a C18 column (*n* = 3), performed in triplicate. (**a**) PCA of metabolomic profiles obtained as a function of sample-to-solvent ratio. PCA score plots demonstrating extracted faecal metabolites between different ratios. Discrimination between extraction solvents 1:5 (dark blue), 1:10 (orange) and 1:20 (light blue) was characterised by a variability of 33.3%. (**b**) A Venn diagram of the mean number of metabolites detected between each method. (**c**) The total number of *m*/*z* features and (**d**) total number of putatively identified metabolites were calculated in positive ionisation mode and (**e**) the overall mean signal intensity of each sample-to-solvent-ratio was assessed. (**f**) The metabolite class quantification demonstrating the faecal metabolome patterns according to chemical class in each extraction sample. The bar chart data were expressed as mean ± SEM and statistical significance was assessed using a one-way ANOVA. * *p* < 0.05 ** *p* < 0.01, *** *p* < 0.001, **** *p* < 0.0001.

**Figure 5 metabolites-12-01110-f005:**
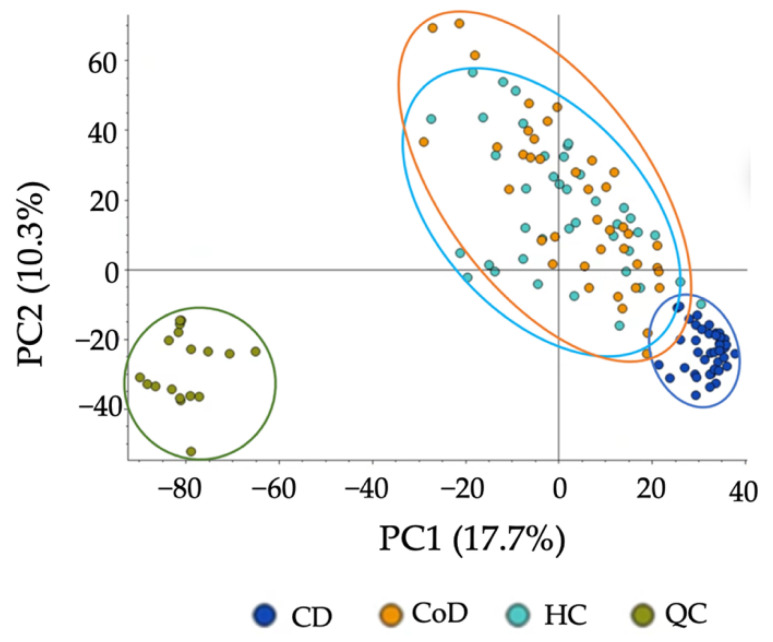
PCA of metabolomic profiles based on untargeted analysis of gastrointestinal disease. PCA score plots demonstrating extracted faecal metabolites between patient groups. Principle Component 1 directionality describes the variance between CD (dark blue), CoD (orange) and HC (light blue) and explains 17.7% of the total variance of the data. QCs are shown in green. The samples were performed in triplicate and are shown as individual datapoints to represent the variance in the dataset.

**Figure 6 metabolites-12-01110-f006:**
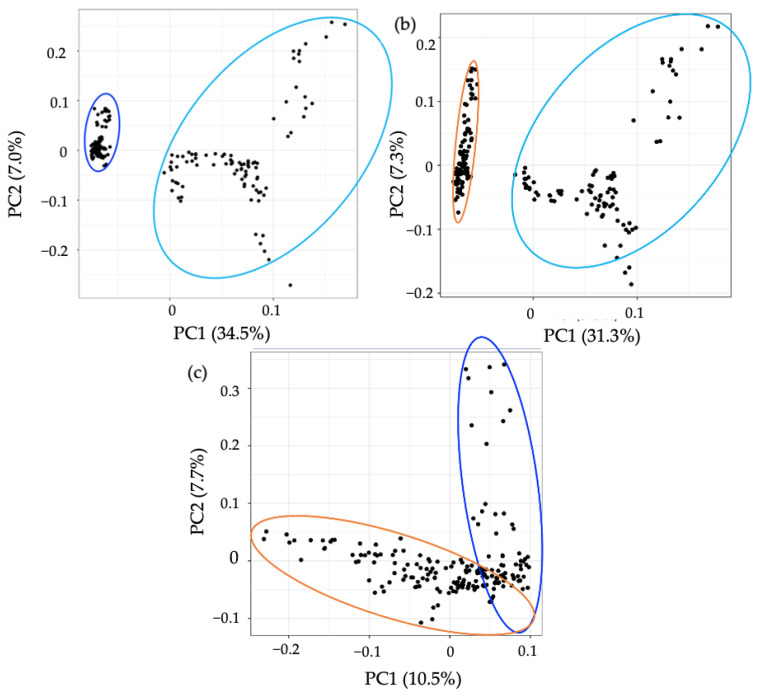
PCA of the metabolomic profiles based on targeted analysis of gastrointestinal disease. PCA score plots demonstrating extracted faecal metabolites between CD (dark blue), CoD (orange) and HC (light blue). The discrimination between (**a**) CD vs. HC, (**b**) CoD vs. HC, and (**c**) CD vs. Co was characterised by variabilities of 34.5%, 31.3%, and 10.5%, respectively. The samples were performed in triplicate and are shown as individual datapoints to represent the variance in the dataset.

**Table 1 metabolites-12-01110-t001:** The details of experimental conditions for each extraction parameter.

Experiment	Independent Variable	Sample Weight	Solvent Used	Cell Lysis Method
1	Sample weight	10 mg, 20 mg, 50 mg, 100 mg	MeOH	Bead beating (5 ms^−1^, 60 s)
2	Extraction solvent	50 mg	MeOH, 1:1 MeOH/H_2_O, 2:1 CHCl_3_/MeOH	Bead beating (5 ms^−1^, 60 s)
3	Cell lysis method	50 mg	MeOH	Bead beating (5 ms^−1^, 60 s), sonication (40 kHz) freeze-thaw cycle (24 h)
4	Sample-to-solvent ratio	50 mg	MeOH	Bead beating (5 ms^−1^, 60 s)

**Table 2 metabolites-12-01110-t002:** Table of patient demographics.

Variable	HC *n* = 20	CD *n* = 20	CoD *n* = 20
Gender			
Female (%)	45	40	60
Male (%)	55	60	40
Age (range)	6.6 (2.3–13.7)	12.3 (7.6–14.8)	9.2 (4.0–14.8)
BMI z-score	0.3	−0.7	0.2

HC, healthy control; CD, Crohn’s disease, CoD, Coeliac disease.

## Data Availability

The data that support the findings of this study are available from the corresponding author upon reasonable request due to privacy or ethical restrictions.

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
