# Peer review of "An Optimised Monophasic Faecal Extraction Method for LC-MS Analysis and Its Application in Gastrointestinal Disease"

_metabolites, 2022, doi:10.3390/metabo12111110_

Round 1

Reviewer 1 Report

Abstract

On line 17 it is mentioned that “a standardized method for extracting metabolites from faecal samples for large-scale metabolomics is yet to be defined”, but this is debatable/subjective. There are other optimized and furthermore also validated methods for extraction of metabolites from faecal samples. How do the authors define ‘standardized’? Please rephrase.

Materials & Methods

·      Line 115: Please clarify how samples were stored after collection and prior to transport. Were samples taken at home or in a hospital setting? Were they frozen immediately or transported fresh?

·      Line 139: Was the LC-MS method optimized (and validated)? Were all settings transferred from an already existing method? Please clarify and/or add appropriate references. It is unclear whether the LC-MS method is fit-for-purpose.

·      Line 157 and 191: Please clarify how identification of targeted metabolites was performed. Did the authors make use of analytical standards?

·      Line 195: please rephrase “For untargeted analysis, multivariate statistical analysis was performed using Compound Discoverer software 3.2” since this software may indeed have been used in preparation of the multivariate statistical analyses, but not the actual multivariate statistical analysis.

·      Line 206: this approach does not allow full identification of metabolites. Please see Schrimpe-Rutledge et al, JASMS 2016.

Results

·      Line 225: How/why did the authors conclude that “metabolite extraction from 225 10 mg and 100 mg samples were unsuitable for metabolomic analysis”?

·      Section 3.1 can be reduced significantly as these findings are to be expected and fairly straightforward. In line with this, the added value of figure 1a, 1b, 1d and 1f is limited. Suggestion to reduce figure 1 by deleting 1a, 1b, 1d and 1f (or move to supplementary). Regarding figure 1d, it should also be noted that these metabolites were not identified strictly speaking.

·      Was identification and classification of metabolites based on m/z solely? Unfortunately, a matching m/z does not suffice for identification.

·      Section 3.2: the added value of figure 2a, 2d and 2f is limited. Suggestion to reduce figure 2 by deleting 2a, 2d and 2f (or move to supplementary). Regarding figure 2d, it should also be noted that these metabolites were not identified strictly speaking.

·      Section 3.3: the added value of figure 3a, 3d and 3f is limited. Suggestion to reduce figure 3 by deleting 3a, 3d and 3f (or move to supplementary). Regarding figure 3d, it should also be noted that these metabolites were not identified strictly speaking.

·      Section 3.4: the added value of figure 4a, 4d and 4f is limited. Suggestion to reduce figure 4 by deleting 4a, 4d and 4f (or move to supplementary). Regarding figure 4d, it should also be noted that these metabolites were not identified strictly speaking.

·      Section 3.5 and Figure 5: please remove or alter since PCA is a type of unsupervised analysis, commonly used to assess variation within the data, but not to describe/model relationships. Please also remove sample replicates when using this approach as only unique samples should be used for this type of analysis.

·      Line 477: please clarify in which way these metabolites were identified, and please note that a match based on accurate m/z does not suffice for confident compound identification.

·      In my opinion, the central network does not have much added value. A simple table would do the trick. 

Discussion

·      Line 506: how can the authors assure that there was no loss of data quality?

·      The added value of figure 8 is limited.

·      Line 529: please rephrase ‘instrument damage’.

·      Line 532: the authors refer to ‘experiment 4’, but there is no mention of ‘experiment 4’ earlier on.

·      Line 523-533: the (failed) extraction of 10 mg and 100 mg should already be mentioned in the results section.

·      Line 603-607: here it is mentioned that the use of standards is foreseen in future work, again that indeed compounds were not confidently identified and classified in the current work. This should be stated more clearly as well as more early on and throughout the paper since this is a very important limitation, thwarting publication.

·      Line 609-616: these claims cannot be made based on the work presented in this paper.

·      Line 618 and onwards: the authors should more in depth discuss (or at least mention) the potential trade-off that is made regarding metabolome coverage. A simple, fast and scalable extractions is not very useful if metabolome coverage is significantly reduced. Please rewrite this section accordingly.

·      Line 631: please support this statement using appropriate references.

·      Line 639: How did the authors assess (improvement of) the quality of data? Please be more specific. 

·      Line 634-653: please support statements with appropriate references.

·      Overall, limitations and contradicting findings should also be mentioned in the discussion, with appropriate references (no cherry picking!). A good discussion requires critical reflection.

Supplementary figures and tables:

·      There are a lot of supplementary figures and tables, and I think the added value of e.g. figure S1 S2, and S3 is limited since these figures do not contain a lot of additional info compared to the manuscript text. By e.g. adding one extra sentence in the text (per figure), these figures could be deleted. Please re-assess the need for alle these figures and tables.

·      For all tables and figures: please specify whether ‘metabolites’ were detected/identified using the targeted or untargeted approach in the caption.

General remarks:

·      The authors state that there is a lack of “standardized” faecal metabolomics extraction and analysis methods useable in large-scale studies. When performing a more more thorough review of available literature however, it should become apparent that this is in fact not the case… Several interesting/important papers on this topic are not mentioned or cited, so maybe the authors may have missed this and/or should further clarify their view on the matter. 

·      Please be prudent regarding the use of the term ‘identified’, since there are different levels of identification (cfr. Schrimpe-Rutledge et al, JASMS 2016). Please take this into account and alter the use of this term throughout the manuscript.

·      It appears all metabolites were annotated based on m/z only? This implies none of the metabolites were in fact identified with sufficient confidence! Identification and classification cannot be claimed.  

The authors stress the need for a high-throughput metabolomics approach, but perform targeted and untargeted metabolomics using two different instruments. Why was targeted and untargeted metabolomics not performed using the same instrument? The use of two separate analyses does not align with the aim of easy or ‘less-complex’ implementation in large scale clinical studies. It could be argued that one single analysis of a sample that underwent biphasic extraction is more high-throughput compared to two analyses of one sample that underwent monophasic extraction. Can the authors better substantiate their claims? Can the targeted and untargeted LC-MS methods be used stand-alone or are both required? 

Also, is the metabolome coverage as good using a monophasic extraction versus biphasic extraction? This may be arguable considering the large physicochemical variation within the metabolome. 

·      In my opinion, assessment of PCA plots does not have added value in section 3.1 to 3.4.

·      The lack of method validation is an important limitation, which should be addressed. Linearity, repeatability, recovery, etc. are not assessed.

·      Since “multivariate analysis” refers to PCA analysis in this work, please always refer to PCA. This is more specific and correct.

Reviewer 2 Report

Manuscript title: An Optimised Monophasic Faecal Extraction Method for LC-MS Analysis and its Application in Gastrointestinal Disease

This is nicely done study.

-- Overall all the methods were performed very well and based on the PCA plots I can say that the chromatography has been good considering the fact that we get a clear separation of the different groups (indicating methods, solvents).  

-- I would like to see a figure (may be in supp material ), a PCA plot with the QC samples included.  

-- The little niggle I have is the number of identified metabolites. The number of identified metabolites looks little high to me.  

This could primarily due to the minimum peak intensity setting used for compound discoverer.  Based on the compound discoverer recommended minimum peak intensity, 

1. Q exactive should set to: 500,000 to 100,000  

2. Orbitrap fusion, lumos and ID-X: 50,000 to 100,000

3. Exactive, Exactive plus, velos; 100,000 to 500,000

4. LTQ orbitrap XL, velos: 25,000 to 100,000

-- The intensity of 10,000 you set (as shown in the supporting info Table S4) is very low.  This could potentially takes days to run the raw files and and because you set the intensity so low, there will be quite a lot of unspecified and false positive hits in the list.

-- the data shown in Fig 1(C,D); Fig 2D; fig 3D they all seems very high to me.  Was any post processing done on the identified metabolites ? 

-- how did you deal with the duplicate identifications ? Was them curated ? Was the final number shown in these figures (shown above) were unique metabolites ? 

-- The Detect Compounds node uses the peak quality information to group the XIC traces for common isotopes of C, H, N, O, and S and optionally for Cl and Br. It ignores XIC traces with low‑quality chromatographic peaks for isotope grouping. By setting the peak rating to 0, you are effectively allowing all the low quality chromatographic peaks for subsequent MS/MS or MS matching.  This is the reason you have very high number of identification.  

-- Since you have used CD v3.2, this version of software has the peak rating scoring system but absent from previous versions.  

This is the meaning of peak rating and its color coding. 

 Gray   N/A

 Orange 0 to 2.5

 Yellow >2.5 to 5.0

 Yellow green >5.0 to 7.5

 Green > 7.5 to 10.0 

Ideally you should have set the minimum peak intensity higher at least 50,000 to 100,000 or you should have set the peak rating to minimally yellow green ie, > 5.  That could have filtered out several spurious identifications.  

-- there will be several instances, out of list of molecules you identified may be 30% will have mzCloud match (> 60), ie, they were matched with MS1 accurate mass and tandem MS/MS spectra match with mzCloud library.  Rest of identifications against the BioCyc, CheBI, Fecal Metabolome Databse, Food and Agriculture Organisation, Human Metabolome Database, KEGG are accurate mass identification only.  How confidence are we about these identifications ?

I do not have any other comments as all the methodologies are applied correctly but I feel the metabolite identification in compound discoverer especially the data processing, noise filtering is little sketchy that leads to high number of spurious identifications.  

Reviewer 3 Report

The manuscript under review is devoted to development of a method for metabolites extraction from faecal samples for subsequent LC-MS analysis. The manuscript is written well and illustrated with corresponding figures and diagrams. Having considered the submission, I have following comments and questions.

1) Page 3, Table 1: What was the volume ratio of solvents in the mixtures of W/MeOH and MeOH/CHCl3?

2) Page 4, line 146: There should be 99.9% instead of 99.99%.

3) Section 2.7, line 162: It is necessary not only to provide the list of metabolites but also to include the Q1/Q3 transitions and other device-specific parameters such as collision energy, declustering potential etc. which are necessary to repeat the experiment.

4) To the sections 2.6 and 2.7: Why  were different columns used for untargeted and targeted screening?

5) Section 3.2: Again, the ratio between the volumes of W/M and M/CHCl3 should be mentioned here. Also, in some literature I could find that the mixture of W/M (1:1) is better for feces extraction and following metabolomics analysis (e.g. see article at 10.3390/molecules26144111). Why was ACN and its mixtures with water and/or ChCl3 not tested as a solvent for metabolites extraction in the experiments?

6) Figure 2 and all related: The color legend representing extraction solvents, volumes etc. is better to be given under the figure or somewhere else to show that it is related to the whole figure but not only to its part (b).

7) Page 16, lines 509-514: I can't imagine using lyophilization of the sample containing pure methanol. This solvent has a melting point at -97.6C, so, does it mean that the sample should be frozen with liquid nitrogen? If so, then this step should be added to the protocol. If the sample was not frozen, then this method is not lyophilization.

8) Fig. 8: Actually, this figure can be moved to Supplementary materials because the corresponding protocol is provided.

9) As I mention above, it is not clear why ACN was neither tested nor discussed as a possible solvent for the extraction of metabolites. I suggest therefore performing additional experiment(s) with his solvent.  

Round 2

Reviewer 1 Report

Thank you for adding info on sample collection and methodology, as well as further clarification regarding compound identification and study limitations. However, please also nuance “A cross-study analysis was performed against metabolites previously implicated in IBD pathophysiology throughout the literature, where it was found that the present method detected 75% of the metabolites in the IBD target list (Supplementary Information Figure S11)” considering this.

Regarding figure 5, the figure caption says “Samples are represented by individual datapoints, performed in triplicate (n=20).” This means not every data point is unique in this PCA. Hence the question to remove sample replicates, which has not been addressed yet.

There are some typos in the newly added/altered text.

Reviewer 2 Report

Thank you for incorporating the changes I have suggested and authors would agree that the data now looks realistic.  

1. Further I would like the authors to provide a supplemental material in form of excel output from the entire CD experiment.  I specifically want to see how many metabolites were actually identified with correct MS/MS mzcloud assignment and how many were matched with only accurate mass matching within 5 ppm error.  Based one the these matching authors can provide an additional column in that supporting info sheet with MSI level of each metabolite match.  If any metabolite has precursor mz match alognwith correct MS/MS assignment and mzclould score > 60 would be MSI level 2.  Metabolites that have accurate precursor ion match but no mzCloud match (which means they were matched to KEGG/Chemspider/Pubchem/Metabolika pathway only) such metabolites can be categorized as MSI Level3.  Thus provide a small table that would reflect this information.  

2. For the targeted analysis, I would like the authors to provide an additional information in Table S3;  Molecular Formula, precursor m/z and product m/z with Retention time information.  

Reviewer 3 Report

Thanks for providing response on all my comments. Please note that for a reviewer it is much more convenient to find any changes in the text when they are highlighted. Moreover, this is a strict recommendation which is always given by the Editor when a revision is requested.

I still have minor recommendations for the authors to improve the text

1) Page 4, Table 2: The abbreviations (HC, CD and CoD) are better to be placed under the table as footnotes.

2) Page 15, line 500: Please correct the text in the same way as it was explained in the rebuttal letter: "samples were evaporated to dryness under vacuum."

Good luck with your article. 
